

# Brief Communication: The global signature of post-1900 land ice wastage on vertical land motion

Riccardo E. M. Riva[1], Thomas Frederikse[1], Matt A. King[2], Ben Marzeion[3], Michiel van den Broeke[4]

[1]Department Geoscience and Remote Sensing, Delft University of Technology, Delft, 2618 CN, The Netherlands

[2]Surveying and Spatial Sciences, School of Land and Food, University of Tasmania, Hobart, Tasmania, Australia

[3]Institute of Geography, University of Bremen, Bremen, Germany

[4]Institute for Marine and Atmospheric Research Utrecht, Utrecht University, Utrecht, The Netherlands

*Correspondence to*: Riccardo Riva (r.e.m.riva@tudelft.nl)

**Abstract.** Melting glaciers, ice caps and ice sheets have made an important contribution to sea-level rise through the last century. Self-attraction and loading effects driven by shrinking ice masses cause a spatially-varying redistribution of ocean waters that affects reconstructions of past sea level from sparse observations. We model the solid earth response to ice mass changes and find significant vertical deformation signals over large continental areas. We show how deformation rates have been strongly varying through the last century, which implies that they should be properly modelled before interpreting and
extrapolating recent observations of vertical land motion and sea level change.

## 1 Introduction

The amount of ice stored on land has strongly declined during the 20[th] century, and melt rates showed a significant acceleration over the last two decades. Land ice wastage is well known to be one of the main drivers of global mean sea-
level rise, as widely discussed in the literature and reflected in the last assessment report of the IPCC (Church et al., 2013). They show that the century-long trend is mainly due to melting mountain glaciers, while the recent acceleration is mostly driven by increased mass loss from the Greenland and the Antarctic ice sheets.

A less obvious effect of melting land ice is the response of the solid earth to mass redistribution on its surface, which, in the first approximation, results in land uplift where the load reduces (e.g., close to the meltwater sources) and land subsidence
where the load increases (e.g., under the rising oceans). This effect is nowadays well known within the cryospheric and sea level communities (Watson et al., 2015). However, what is often not realized is that the solid earth response is a truly global effect: a localized mass change does cause a large deformation signal in its proximity, but also causes a change of the position of every other point on the Earth's surface. The theory of the Earth's elastic response to changing surface loads forms the basis of the 'sea-level equation' (Farrell and Clark, 1976), which allows sea-level fingerprints of continental mass
change to be computed.



In this brief communication, we provide the first dedicated analysis of global vertical land motion driven by land ice wastage. By means of established techniques to compute the solid earth elastic response to surface load changes and the most recent datasets of glacier and ice sheet mass change, we show that land ice loss currently leads to vertical deformation rates of several tenths of mm per year at mid-latitudes, especially over the Northern Hemisphere where most sources are located.

In combination with the improved accuracy of space geodetic techniques (e.g., Global Navigation Satellite Systems), this means that the effect of ice melt is non-negligible over a large part of the continents. In particular, we show how the recent acceleration in melt rates affects estimates of secular vertical land motion, and therewith has an impact on various geodetic applications, including estimates of long-term sea level rise at tide gauges. While elastic deformation of the earth has been widely considered due to especially atmospheric loading changes, the effects of ice loading changes have been largely

ignored, due to the difficulty of an accurate quantification (Santamaria-Gomez and Memin, 2015).

## 2 Datasets and methods

As in Frederikse et al. (2016), we consider yearly mass losses from glaciers and ice caps and the Greenland and Antarctic ice sheets. For glacier mass loss, the recent estimate of Marzeion et al. (2015) is used. For the Greenland ice sheet during the

period 1902-1992 we use data from Kjeldsen et al. (2016). Between years 1993-2014, we use an input-output approach for both Greenland and Antarctica: input is based on the modelled RACMO2.3 surface mass balance (Van den Broeke et al., 2016); Greenland ice discharge is also based on van den Broeke et al. (2016). For Antarctica, the ice discharge is parametrized as a constant acceleration of 3.0 Gt/y$^2$, starting from equilibrium between 1902-1992, which gives a good fit with IMBIE estimates (Shepherd et al., 2012). Figure 1 shows the location of the glaciers and ice sheets on which our mass

balance estimates are based, together with the cumulative mass loss of the individual iced regions. The glacier mass loss is regionalized following the regions described in Pfeffer et al. (2014). For both ice sheets, the spatial distribution of the mass change is based on linear trends obtained from GRACE JPL mascon solutions (Watkins et al., 2015), scaled to match the estimated total mass loss (Frederikse et al., 2016). Note that this approach will bias the resulting fingerprints towards post-2002 values (e.g., the signal over the Antarctic Peninsula contains the signature of the glacier acceleration following the

2002 Larsen B ice shelf breakup), but with a limited effect on the far-field signal.

We determine the solid earth elastic response by solving the sea level equation (Farrell and Clark, 1976) for each load at each year and add them together to obtain the total response. We follow a pseudo-spectral approach (Tamisiea et al., 2010) in the centre-of-mass of the earth sytem (CM), solved up to spherical harmonic degree 360 for a compressible earth, including the effect of induced changes in the earth's rotation. We then estimate a linear trend over the period of interest

through the resulting vertical land motion time series by means of ordinary least squares.





**3 Results**

Global maps of vertical deformation rates are shown in Figure 2 for different time spans. In all panels, the largest values are reached at the location of the melt sources, while the far-field negative deformation is shaped by the change in the position of the earth rotation axis (the so-called "solid earth pole tide").

The near-field deformation rates are dominated by the direct effect of the individual melt sources. The maximum uplift rates

range from about 4 mm/yr (panel a) to 11 mm/yr (panel d), though the exact values are dependent on the model resolution (0.5 degrees) and by the accuracy of the melt distribution.

In the far field, here loosely defined as regions located several hundred kilometers away from any region of ice mass loss and characterized by small gradients in vertical deformation rates, maximum uplift rates have increased from less than 0.6 mm/yr over the last century to about 1.0 mm/yr during the last decade. Larger rates are combined with a southward shift of the 0.4

mm/yr contour, which has moved over North America from South of Hudson Bay to about Washington D.C., and over Europe from Denmark to Northern Italy. During the last decade, most of Australia has been subsiding at rates larger than 0.4 mm/yr. Interestingly, the far-field deformation pattern in Central Asia is not very different through the century, being dominated by the relatively constant effect of glacier mass loss on and around the Tibetan Plateau.

In order to better show the temporal evolution of vertical land motion through the last century, in Figure 3 we display time series of the signal and of its time derivative for six major cities worldwide. Note that deformation rates have been computed after using a 15-year moving average. We have specifically chosen coastal cities because they are representative of the far-field deformation over large portions of the continents, due to the smoothness of the signal, but also because vertical land motion has a direct effect on sea level change and on tide gauge measurements of that change.

In the course of the 113 years covered by this study, cities in the Northern Hemisphere have accumulated several centimeters of uplift (2.8 cm for New York, 3.9 cm for London, and 5.0 cm for Seattle), while cities in the Southern Hemisphere have subsided (Rio by 1.0 cm, Sydney by 3.4 cm). At lower latitudes the signal is smaller, e.g. Shanghai has been uplifted by 1.0 cm.

The vertical motion has not been constant in time, following temporal variations in ice mass loss rates. In particular, rates

were lower at the beginning of the last century and in the 1970's, while a clear acceleration can be seen during the last 20 years, when the ice sheets contribution has increased. Interestingly, the recent high rates are not exceptional at all locations, depending on the relative distance from specific glaciers and the two ice sheets. For example, in Seattle rates above 0.6 mm/yr have already been reached in the 1930's, while in London the recent rates of about 0.5 mm/yr are lower than those experienced in the 1930's. The increased contribution of the ice sheets can also lead to a reduction in the local deformation

rates; this is the case for Shanghai, which is currently experiencing very little vertical motion associated with ice melt, due to its location on the transition line between uplift driven by northern sources and subsidence enhanced by the small recent mass gain in East Antarctica.




## 4 Discussion

Long-term vertical land motion in the near-field is dominated by the effect of ice loss, which allows geodetic observations to be used to quantify ice mass change (e.g., Bevis et al., 2012) or to separate the effect of present-day mass loss from the signature of glacial isostatic adjustment (GIA) (e.g., Kahn et al., 2016).

In the far field, several competing processes can lead to inter-annual vertical displacement rates at the millimeter-per-year level. For this reason, geodetic observations are usually corrected for the effect of a number of loading processes related to

water mass redistribution, such as changes in atmospheric pressure, land hydrology and ocean mass (Santamaria-Gomez and Memin, 2015). This approach is problematic when studying the effect of climate change, since current models of the water cycle are not accurate in terms of secular variations, which are orders of magnitude smaller than those driven by the seasonal cycle.

The remaining signal is usually attributed to geodynamic processes (e.g., GIA, earthquakes, volcanoes, landslides) or to local

effects, either of natural (e.g., ground compaction, sediment transport) or of anthropogenic origin (e.g., groundwater and hydrocarbons extraction, dam building). Of all those processes, only GIA can currently be modeled globally (e.g., Peltier et al., 2015), albeit with large uncertainties. While the pole tide effect is modelled in geodetic data analysis, it is entirely limited to quasi-annual variations (King and Watson, 2014) meaning that the signal identified herein is not considered.

The general approach in sea level studies until early 2000's has been to neglect any non-GIA signal, meaning that sea level

estimates based on tide gauges have been potentially biased by several tenths of millimeter per year, as recently discussed by Hamlington et al. (2016). More recently, estimates of vertical land motion have been obtained by the combination of observations from satellite altimetry and tide gauges (Nerem and Mitchum, 2002) or by direct observations by means of GPS (e.g., Woppelmann and Marcos, 2016). However, those approaches are limited by the fact that space geodetic observations are only available since the 1990's, when the effect of ice wastage has been considerably larger than during the rest of the

20$^{th}$ century (Fig. 2c vs. Fig. 2b). The assumption of constant rates throughout the century means that significant errors in sea level reconstructions based on tide gauge records will still be present, even after correcting for vertical land motion as observed by GPS.

Global vertical land motion is also changing the shape of the ocean basins (Fig.2), which introduces a bias in sea level change estimates based on satellite altimetry. However, during 1993-2014, this effect is less than -0.11 mm/yr over the

global ocean (-0.07 mm/yr between +/- 66 degrees latitude), largely within the uncertainty of space-based estimates of global mean sea level change. Additionally, the vertical land motion discussed in the GPS context could induce an additional bias in altimetry estimates, due to the use of tide gauges to determine altimeter drift (Watson et al., 2015).

It is worth noticing that we have modelled the earth's elastic deformation, but neglected the viscoelastic response of the mantle (Farrell and Clark, 1976; Peltier et al., 2015); however, the far-field signature of relaxation is controlled by bulk

viscosity values, which are expected to provide a significant response at time scales much longer than those covered by this study.



Considering the recent improvement in mass loss reconstructions of glaciers and ice caps (Marzeion et al., 2015) and ice sheet (Shepherd et al., 2012) mass loss, even though the 20[th] century contribution of Antarctica is still poorly understood, we advocate direct modelling of the effect of time-varying ice wastage as a way to improve the accuracy of sea level change

estimates (Frederikse et al., 2016).

### 5 Conclusions

We have shown how land ice wastage through the last century has caused vertical land motion in the order of several tenths of mm per year over large parts of the continents. Deformation rates are highly non-linear and location dependent, with

larger values between 1930-1950, minima around 1970 and a clear acceleration during the last two decades.

This effect is particularly important in the context of sea level studies, since several of the longest tide gauge records are at mid-latitudes in the Northern Hemisphere, where the effect of melt of Arctic glaciers and the Greenland ice sheet is large, as also discussed by Thompson et al. (2016).

In particular, due to the recent acceleration in land ice melt, it should not be assumed that linear rates estimated by GPS over

the last two decades are representative of centennial vertical land motion, without first accounting for the effect of glacial mass change.

### Data policy

The data used to generate Fig.2 and Fig.3, in the form of NetCDF files containing gridded values of annual vertical

deformation, are freely available through the 4TU.Centre for Research Data at http://data.4tu.nl [*exact link to be published after acceptance of the manuscript*].

For the data used to generate Fig.1, we refer to the original papers.

### Author contribution

REMR and TF designed the study; TF performed the computations and produced the figures; REMR wrote the paper; MAK contributed to the analysis of the results; BM provided glacier mass balance data; MvdB provided ice sheet SMB data; all authors commented on the manuscript.

### Acknowledgments

REMR and TF acknowledge funding from The Netherlands Organisation for Scientific Research (NWO) through VIDI grant no 864.12.012. MAK is a recipient of an Australian Research Council Future Fellowship (project number FT110100207) and supported by Australian Research Council Special Research Initiative for Antarctic Gateway Partnership (Project ID SR140300001). MvdB acknowledges funding from the Netherlands Polar Program and the Netherlands earth System Science Center (NESSC). We are grateful to Kristian Kjeldsen en Kurt Kjær for sharing Greenland mass loss data.




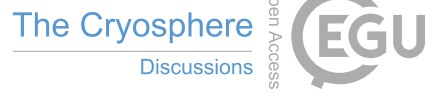

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

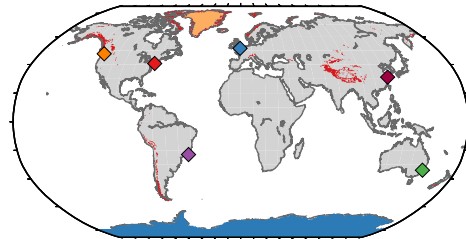
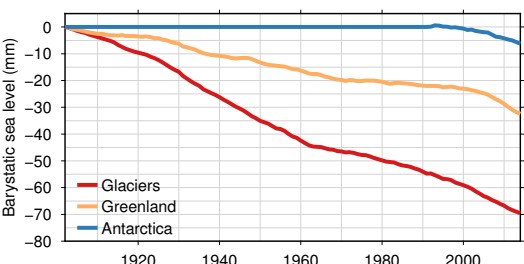

**Figure 1: Left: Land ice regions considered in this study (red, glaciers; orange, Greenland; blue, Antarctica), with colored diamonds representing the coastal cities of Fig.3. Right: global mean sea level contribution of ice wastage between years 1902-2014.**




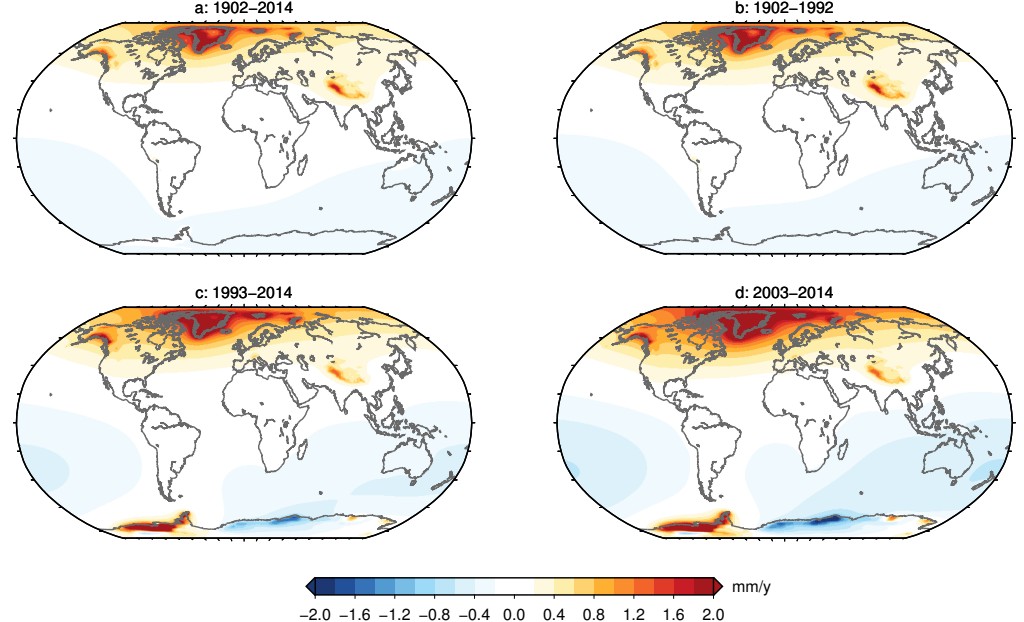

**Figure 2: Maps of average vertical deformation rates over different time spans. a: full time span covered by this study; b: pre-satellite era; c: the GPS era; d: the GRACE era.**

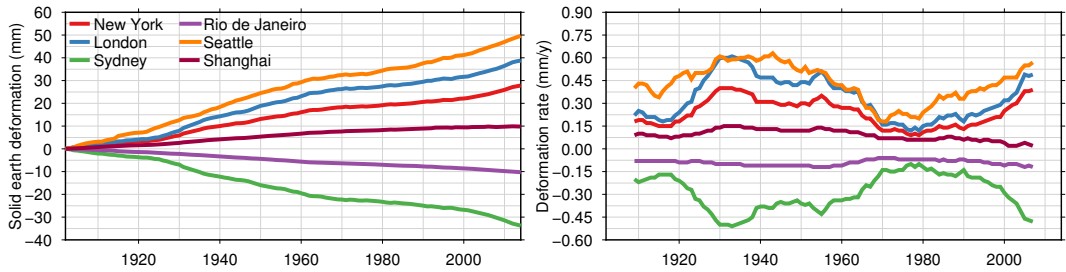


**Figure 3: Time series of vertical deformation (left panel) and 15-year-average rates (right panel) at selected coastal cities due to global ice mass changes. The locations of the cities are indicated by diamonds in Fig.1a, following the same color-coding.**