# Peer review of "Brief Communication: The global signature of post-1900 land ice wastage on vertical land motion"

_The Cryosphere, 2016_

## Referee Comment (RC1) · A. Santamaría-Gómez (Referee) · 5 Jan 2017

General comments:

This paper addresses the elastic deformation of the Earth's crust in the vertical component due to ice mass changes during the last century. This topic is of great relevance for the sea-level community, where vertical land motion (VLM) corrections of the tide gauge (TG) records are needed in order to observe climate-driven sea-level change. In particular, this paper emphasizes the non-linear vertical deformation in far-field areas where most of the TGs used for sea-level change are installed. VLM corrections at the TGs are typically obtained from GIA crustal uplift predictions and, more recently, also from space geodetic observations (GPS). Despite the different advocacy for GIA or

GPS corrections found in the literature, both share a common limitation: they assume VLM at the TGs is linear throughout the sea-level record. This assumption is understandable due to the lack of independent observations to better constraint the VLM at the TGs. This paper provides new insights on the level of non-linear VLM expected from recent ice mass changes and completes the assessment of the non-linear VLM bounds from atm, ocean and hydrology loading. Therefore I recommend this paper for publication after considering some minor comments below.

Specific comments:

L21: "the century-long trend" in ice-mass loss . . . Also, a reference to the Fig. 1 (right) would be appropriate.

L26: "what is often not realized" by who? I believe is quite common to deal with solid Earth deformation due to loading at global scale.

L56: while the secular or mean VLM trends are probably indistinguishable in a CM or CE/CF frame, the interdecadal vertical deformation may be different depending on the chosen frame, which, in turn, may have an impact on the short-term trends shown in figs. 2 and 3. This is what happens with other loadings (atm, ocean and hydro) at the interannual variations leaving the long-term trend unchanged. Maybe it does not happen with the spatial pattern of the ice-mass unloading, so I suggest adding a sentence explaining why the CM frame was chosen and whether it has any impact on the results.

Fig. 2: if the format of the communication allows it, I would suggest to add two more maps showing the rate differences between the maps a) and c) and a) and d). This would support the discussion of the results and also fig. 3.

L66: accuracy of both, the melt distribution and the regional mass loss values.

L71: "most of Australia has been subsiding at rates larger than 0.4 mm/yr" this has been observed by GPS estimates since long ago without any plausible explanation

thus far (see for instance Altamimi et al 2016). I suggest emphasizing this point.

L71: This is a very interesting spatial pattern in which northern TGs are uplifted faster in the last decade (captured by the GPS VLM corrections) compared to the last century, while southern TGs have subsided faster. This could partially explain the hemispheric difference in sea-level rise found by Wöppelamn et al. 2014. At the time that paper was published, this ice-mass loss fingerprint was unknown and it seems to me from your Fig. 2 that the average difference between the northern and southern TGs used by Wöppelman et al. 2014 could accommodate part of the hemispheric difference that was not explained by the uncertainties.

L71: In relation to my comments above. Similar to the GIA effect on the deepening of the ocean basins and the resulting global mean sea-level change (of about 0.3 mm/yr), is there any ocean basin effect due to recent ice-mass loss to be accounted for in the sea-level trend?

L89: The estimated changes in VLM rates appear to induce a periodic-like oscillation close to 60 years, especially in northern TGs close to the areas of ice-mass loss. Many of these TGs have very long records and were used to assess a global 60-year oscillation in sea-level by Chambers et al. 2012. I wonder how much of the observed 60-year oscillation is due to the ice-mass loss fingerprints shown here. A detailed analysis would be worth pursuing. A priori, the oscillation phase shown by Chambers et al. 2012 (Fig. 1) is consistent with your results.

L99-101: Note that we didn't correct or encouraged correcting for continental water mass loading due to the significant differences amongst the model outputs in terms of secular, as you mention in the next sentence, but also interannual deformation.

L113: "those approaches are limited by the fact that space geodetic observations are only available since the 1990's". Note that there exist alternative approaches in combining satellite altimetry and tide gauge observations that benefit from the longer TG series, thus reducing this limitation (see for instance Kuo et al., 2004 and Santamaría-

[Figure]

Gómez et al. 2014).

L115-117: This is probably the biggest limitation of using GPS for correcting long TG records (together with the lack of nearby GPS observations), especially when very short GPS series are used. However, it is not a limitation exclusive of the GPS VLM corrections, but also when using GIA corrections which neglect any non-linear VLM in addition to any other linear VLM that is not GIA.

L117: In relation to my comment above. The average VLM for the last 10 years for the 6 TGs shown in Fig. 3, does not seem to lie far from the average VLM over the last century. It would be interesting to have some statistics of the VLM deviation during the GPS era or the additional maps I suggested above.

L140: This is an interesting perspective, but one also needs to consider the uncertainties in the ice-mass loss fingerprints, which were not discussed in this brief communication. In addition, even after correcting for this effect, the VLM corrections (from GPS or GIA) will still be considered linear as a working hypothesis even if we have clues that they may not be (due to pole motion deformation, hydrologic loading, long-memory noise, etc.).

Technical corrections:

L28: "position of every other point on the Earth's surface" with respect to the Earth's center of mass.

L48: "cumulative mass loss" should be "equivalent sea-level change" or "barystatic sea-level change".

L121: "induce" I would suggest "reveal" here.

---

## Referee Comment (RC2) · Anonymous Referee #2 · 20 Jan 2017

General comment

This note concerns the effects of continental ice melting on vertical displacement in the time-window from 1900 to present. It is a timely study, which provides an important contribution to the understanding of long-term vertical movements of the Earth's crust, a topic of interest for the cryospheric and geodesy communities. I have a few comments, listed below.

Specific comments

Abstract. "Deformation" should be replaced by "vertical displacement" here and in the rest of the paper. They are used as synonyms but they are not, in my opinion.

Line 23. Another less obvious effect that could be mentioned is the variation of gravitational potential \Phi that together with U give relative sea-level change according to the sea-level equation S=\Phi/\gamma + c -U where c is the notorious c-constant.

L26. I think this is realised, indeed, also in the cryospheric community.

L28. Actually the SLE is more general and can also deal with the viscoelastic Earth's response.

L30. In this brief communication... From what I have understood, the novelty here is the long time window considered (1900-now) for the computation of the elastic displacement, and the use of realistic ice sources.

L40. Quantification is not so problematic if the melting histories are well constrained.

L58. Adding the individual responses to obtain the total response is OK if you assume linearity. An indeed the SLE is linear as long as you do not allow for shoreline migration. But I guess that here the shorelines do not move.

L58. Compressible is OK. But I imagine also layered and consistent with the seismic travel times.

L59. 'period of interest' is vague. From the figures I see different rates at different times, that appears to contradict the use of a unique linear trend.

L65ff. It can be worth to recall that these fingerprints have a vanishing global average.

L68. I am not sure that 'pole tide' is appropriate. From e.g., http://www.navipedia.net/index.php/Pole_Tide I understand that the pole tide is related with the 14-months Chandler Wobble, which I am sure the authors have filtered out from their equations. What causes the lobes in the far field in the vertical displacements maps is the (non-oscillatory) secular component of polar motion.

L68ff. Where are these max values met?

[Figure]

L76... Has been subsiding... well, the actual subsidence stems from this component plus GIA, etc. etc

L85. Ditto. See L76. These subsidences are virtual, they only represent one component of total subsidence, and probably not the largest one.

L84. Vertical displacement has certainly an effect on tide gauge. But also N = \Phi/\gamma + c has one. Is this negligible? Has this been computed? In a more in-depth study I recommend to show both S and N along with U, for the same sources considered in this study.

L111. The coseismic displacement can be also modelled globally (see http://onlinelibrary.wiley.com/doi/10.1029/2003GL019347/full).

L112. What is the signal identified therein? Unclear. Is the rate of solar motion driven by the ice sources considered? What is its amplitude and direction?

L130. I do not understand why the 'far field signature' is mentioned here. Viscosity also controls deformation in the near field.

See http://journals.fcla.edu/jcr/article/view/80095/77355 for advice on how hyphenate "sea level".

I remark the importance of providing gridded values of the fields computed here to the community.

---

## Author Comment (AC1) · 31 Mar 2017

**Answers to the reviews of manuscript "Brief Communication: The global signature of post-1900 land ice wastage on vertical land motion" by Riva et al. (2016), doi:10.5194/tc-2016-274.**

We wish to thank the referees for their feedback on our manuscript.
Below we respond to each individual comment, where text by the referees is in bold.

On behalf of all authors,
Riccardo Riva

**Referee #1 (Alvaro Santamaría-Gómez)**

**L21: "the century-long trend" in ice-mass loss . . . Also, a reference to the Fig. 1 (right) would be appropriate.**

Done.
We have added "in ice-mass loss" and a reference to Fig.1

**L26: "what is often not realized" by who? I believe is quite common to deal with solid Earth deformation due to loading at global scale.**

We have changed the sentence into "what those communities often do not realize".
From experience, we have the feeling that outside the geodetic community, to which Reviewer 1 belongs and which routinely deals with loading effects, scientists are mostly aware of the near field effects.

**L56: while the secular or mean VLM trends are probably indistinguishable in a CM or CE/CF frame, the interdecadal vertical deformation may be different depending on the chosen frame, which, in turn, may have an impact on the short-term trends shown in figs. 2 and 3. This is what happens with other loadings (atm, ocean and hydro) at the interannual variations leaving the long-term trend unchanged. Maybe it does not happen with the spatial pattern of the ice-mass unloading, so I suggest adding a sentence explaining why the CM frame was chosen and whether it has any impact on the results.**

We carefully considered whether to present vertical deformation in the Centre-of-Mass of the Earth System (CM) of Centre-of-Figure of the Solid Earth (CF) frame, after having computing both.
What we found is actually the opposite of what has been sketched by the reviewer: secular VLM trends are largely affected by the choice in reference frame, especially in the far-field, while the difference between secular and decadal trends is mostly significant in the near field, which means it is roughly reference-frame independent.
From the point of GPS observations, it would have made sense to use the CF frame, since CM-CF motion is accounted for by the underlying global reference frame (albeit the reference frame realization introduces uncertainties of its own).
However, from the point of sea level research, we believe that it makes more sense to look at vertical land motion in the CM frame, since that is the most natural reference (the sea surface at rest follows the geoid, which is centered at the CM).
We have added an explanatory sentence to the text about why we chose the CM, but we deem the discussion of the impact of the reference frame choice on the modelled signal to be too technical, hence possibly confusing, for the broader TC audience.

**Fig. 2: if the format of the communication allows it, I would suggest to add two more maps showing the rate differences between the maps a) and c) and a) and d). This would support the discussion of the results and also fig. 3.**

We agree with the suggestion and we have added two panels to Fig.2 (below), showing differences between the secular and the decadal trends. The practical need to use the same colour scale for all panels mostly highlights near-field differences, but it is luckily enough to highlight the larger mid-latitude trends in the last decade. This indeed supports the discussion of the time-variable trends in Fig.3, especially for New York, London and Sidney which experience a considerable acceleration in recent times.
We have added a brief discussion of the new panels while describing Fig.2 and a reference to them while discussing the right panel of Fig.3.

[Figure]

Figure 2: Maps of average vertical deformation rates over different time spans. a: full time span covered by this study; b: pre-satellite era; c: the GPS era; d: the GRACE era; e: panels c-a; f: panels d-a.

Note: from this point on there seems to be an offset of 5 lines between the numbering used by the reviewer and the online version of the manuscript (e.g., L66 below should be line 71).

**L66: accuracy of both, the melt distribution and the regional mass loss values.**

Agree, added "and the regional mass loss values".

**L71:"most of Australia has been subsiding at rates larger than 0.4 mm/yr" this has been observed by GPS estimates since long ago without any plausible explanation thus far (see for instance Altamimi et al 2016). I suggest emphasizing this point.**

We have added a sentence to highlight this issue, but we cannot accommodate the suggested reference due to limitations of the Bref Communication format.

**L71: This is a very interesting spatial pattern in which northern TGs are uplifted faster in the last decade (captured by the GPS VLM corrections) compared to the last century, while southern TGs have subsided faster. This could partially explain the hemispheric difference in sea-level rise found by Wöppelamn et al. 2014. At the time that paper was published, this ice-mass loss fingerprint was unknown and it seems to me from your Fig. 2 that the average difference between the northern and southern TGs used by Wöppelman et al. 2014 could accommodate part of the hemispheric difference that was not explained by the uncertainties.**

Wöppelmann et al. (2014) indeed found a hemispheric difference of about 0.9 mm/yr in sea level rise at GPS-corrected tide gauge stations, with larger values in the Northern Hemisphere. From the new panels of Figure 2 (e and f) it can be seen that GPS trends in the last 1-2 decades might overestimate the secular hemispheric difference by more than 0.4 mm/yr (e.g., by comparing New York with Hobart, which in the cited paper show trends close to the corresponding hemispheric means). Indeed, this could potentially explain a large part of the hemispheric difference discussed by Wöppelmann et al. (2014).
However, an exact estimate of this effect would require repeating their experiment by making use of all the 76 tide gauges used in that study. Hence, we have added a comment and the suggested reference in the discussion section (after line 122 in the original manuscript), but not given any hard number on the size of the bias potentially induced by non-linear VLM (simply referred to as "up to a few tenths of mm/yr").

**L71: In relation to my comments above. Similar to the GIA effect on the deepening of the ocean basins and the resulting global mean sea-level change (of about 0.3 mm/yr), is there any ocean basin effect due to recent ice-mass loss to be accounted for in the sea-level trend?**

This point is actually already discussed in the discussion section, at lines 123-126. The effect is about -0.1 mm/yr: noticeable, but within the uncertainty of global mean trends based on, e.g., satellite altimetry.

**L89: The estimated changes in VLM rates appear to induce a periodic-like oscillation close to 60 years, especially in northern TGs close to the areas of ice-mass loss. Many of these TGs have very long records and were used to assess a global 60-year oscillation in sea-level by Chambers et al. 2012. I wonder how much of the observed 60-year oscillation is due to the ice-mass loss fingerprints shown here. A detailed analysis would be worth pursuing. A priori, the oscillation phase shown by Chambers et al. 2012 (Fig. 1) is consistent with your results.**

We thank the reviewer for another suggestion about potential implications of our results.

However, we find it difficult to assess the impact of the VLM variability on the results by Chambers et al. (2012), for at least two reasons: first of all, it is not possible to quantitatively compare VLM with relative sea level changes, since the latter also include the effect of ocean mass changes and geoid changes; secondly, the oscillation found by Chambers et al. (2012) is centered around a zero mean, while our rates remain positive or negative (depending on the hemisphere), since the net cryospheric contribution never changes sign.

Hence, while glacial fingerprints might have modulated long-term oscillations in regional sea level, we prefer not to comment on the issue, considering the impossibility to assess the size of this effect on the basis of VLM fingerprints alone.

**L99-101: Note that we didn't correct or encouraged correcting for continental water mass loading due to the significant differences amongst the model outputs in terms of secular, as you mention in the next sentence, but also interannual deformation.**

We have added that models outputs are also uncertain in terms of interannual signals.

**L113: "those approaches are limited by the fact that space geodetic observations are only available since the 1990's". Note that there exist alternative approaches in combining satellite altimetry and tide gauge observations that benefit from the longer TG series, thus reducing this limitation (see for instance Kuo et al., 2004 and Santamaría-Gómez et al. 2014).**

Thank you for those references, we have added a reference to Santamaría-Gómez et al. (2014) at line 117. Considering that, to our knowledge, those alternative techniques are not yet widely used, we have further edited the sentence at line 118 by writing "the majority of those approaches", instead of "those approaches".

**L115-117: This is probably the biggest limitation of using GPS for correcting long TG records (together with the lack of nearby GPS observations), especially when very short GPS series are used. However, it is not a limitation exclusive of the GPS VLM corrections, but also when using GIA corrections which neglect any non-linear VLM in addition to any other linear VLM that is not GIA.**

True. The fact that using GIA models to correct of VLM does not solve all problems has already been mentioned earlier in the same section. The fact that several processes can induce non-linear VLM has not been mentioned explicitly, simply because those processes are not the object of this study.

**L117: In relation to my comment above. The average VLM for the last 10 years for the 6 TGs shown in Fig. 3, does not seem to lie far from the average VLM over the last century. It would be interesting to have some statistics of the VLM deviation during the GPS era or the additional maps I suggested above.**

We have decided to add two panel to Figure 2, as earlier suggested by the same reviewer. The new plots show that the non-linearity effect in the far-field, where all 6 cities are located, is mostly visible during the last decade.
At line 122 we have added "especially if the observations have been collected during the last decade".

**L140: This is an interesting perspective, but one also needs to consider the uncertainties in the ice-mass loss fingerprints, which were not discussed in this brief communication. In addition, even after correcting for this effect, the VLM corrections (from GPS or GIA) will still be considered linear as a working hypothesis even if we have clues that they may not be (due to pole motion deformation, hydrologic loading, long-memory noise, etc.).**

Indeed, we have not directly assessed uncertainties in the ice mass loss fingerprints, even though those are part of the previous study by Frederikse et al. (2016), on which the fingerprints are based. It is also true that many other unmodelled processes might induce non-linear motions. Nonetheless, it seems reasonable to assume that VLM induced by ice melt currently represents the largest signal at regional scales, and as such should be modelled as well as possible.

We have rephrased the last sentence, which now reads: "In particular, due to the recent acceleration in land ice melt, which represents one of the largest drivers of regional vertical land motion, the estimation of secular rates from GPS observations should account for the effect of glacial mass change".

**Technical corrections:**
**L28: "position of every other point on the Earth's surface" with respect to the Earth's center of mass.**

Actually, changes in surface load will always change the 3D position of every other point at the Earth's surface, while fixing a certain reference frame will determine the size and direction of that change. Considering that our statement is only qualitative, we don't see the need for specifying a reference frame. We have considered adding "with respect to their initial position", but that seemed implicit in the wording "change the position".

**L48: "cumulative mass loss" should be "equivalent sea-level change" or "barystatic sea-level change".**

Agree, we now write "equivalent sea-level change".

**L121: "induce" I would suggest "reveal" here**

We agree that "induce a bias" is possibly not the best phrasing and decided to change it into "cause a bias".

---

## Author Comment (AC2) · 31 Mar 2017

**Answers to the reviews of manuscript "Brief Communication: The global signature of post-1900 land ice wastage on vertical land motion" by Riva et al. (2016), doi:10.5194/tc-2016-274.**

We wish to thank the referees for their feedback on our manuscript.
Below we respond to each individual comment, where text by the referees is in bold.

On behalf of all authors,
Riccardo Riva

**Referee #2**

**Abstract. "Deformation" should be replaced by "vertical displacement" here and in the rest of the paper. They are used as synonyms but they are not, in my opinion.**

We agree that the two words are not synonyms, with "displacement" being a purely kinematic concept especially valid when talking pointwise (displacement is a change in the position of a point or of all points of a rigid object), while "deformation" better refers to the relative motion between sets of points (with the more general meaning of "change of shape"). Hence, we argue that it is appropriate to talk about "GPS measuring displacement" and "ice melt causing deformation". As such, we would rather keep using both words, though we have made an additional effort to use each of them consistently through the paper.

**Line 23. Another less obvious effect that could be mentioned is the variation of gravitational potential \Phi that together with U give relative sea-level change according to the sea-level equation S=\Phi/\gamma + c -U where c is the notorious c-constant.**

True, but maybe confusing, since the paper expressly only deals with vertical land motion.

**L26. I think this is realised, indeed, also in the cryospheric community.**

We are not sure whether the reviewer expects us to remove the sentence, or agrees with our viewpoint. In any case, we admit that it is difficult to quantify which portion of a community is aware of a specific concept. That is why we have originally opted for the wording "what is often not realised", which we believe we can defend based on our personal experience.

**L28. Actually the SLE is more general and can also deal with the viscoelastic Earth's response.**

True. Even though this paper only deals with elastic deformation, it is a good idea to have statements of more general validity in the introduction. We have changed "elastic" into "viscoelastic".

**L30. In this brief communication. . . From what I have understood, the novelty here is the long time window considered (1900-now) for the computation of the elastic displacement, and the use of realistic ice sources.**

Indeed. We now mention the long time window and of the use of realistic ice sources as an additional innovation of this study.

**L40. Quantification is not so problematic if the melting histories are well constrained.**

We did mean an accurate quantification of the melting histories. We now specify it.

**L58. Adding the individual responses to obtain the total response is OK if you assume linearity. An indeed the SLE is linear as long as you do not allow for shoreline migration. But I guess that here the shorelines do not move.**

Indeed. We have added a sentence explaining that our superimposition approach is allowed by the fact that the SLE is linear since we make use of fixed coastlines.

**L58. Compressible is OK. But I imagine also layered and consistent with the seismic travel times.**

Indeed. We now write "compressible and spherically layered". Consistency with seismic travel times is, to our knowledge, standard practice.

**L59. 'period of interest' is vague. From the figures I see different rates at different times, that appears to contradict the use of a unique linear trend.**

We meant to refer to the various time windows shown in Fig.2. We now say "over each time window under study".

**L65ff. It can be worth to recall that these fingerprints have a vanishing global average.**

We are afraid that such a statement will be obvious to people familiar with spherical harmonics, but confusing to many other potential readers. In addition, when sampled at discrete points (e.g., GPS stations or tide gauges) these fingerprints will probably still lead to non-zero global mean values due network geometry issues. Hence, we prefer not to add the suggested comment.

**L68. I am not sure that 'pole tide' is appropriate. From e.g., http://www.navipedia.net/index.php/Pole_Tide I understand that the pole tide is related with the 14-months Chandler Wobble, which I am sure the authors have filtered out from their equations. What causes the lobes in the far field in the vertical displacements maps is the (non-oscillatory) secular component of polar motion.**

We are sure that the terminology "solid earth pole tide" is appropriate. In the given link, the Chandler Wobble is only provided as an example. The term is mostly used within the geodetic community, that's why we had only mentioned it within brackets.
Nonetheless, we have removed the word "pole tide", since we reckon that the terminology may be misleading (the pole tide is not a "regular" tide, in the sense that it originates from Earth's rotation instead of from gravitational attraction by external bodies).
As a consequence, at line 112 we now write "earth rotational effects" instead of "pole tide".

**L68ff. Where are these max values met?**

These max values are met over Greenland, we now specify this in the manuscript.

**L76. . . Has been subsiding. . . well, the actual subsidence stems from this component plus GIA, etc. etc**

True. We thought this was implicit, but it may be better to specify it once more. We have added "because of contemporary ice mass change".

**L85. Ditto. See L76. These subsidences are virtual, they only represent one component of total subsidence, and probably not the largest one.**

Same as above. We now specify "due to continental ice mass loss, cities...".

**L84. Vertical displacement has certainly an effect on tide gauge. But also N =\Phi/\gamma + c has one. Is this negligible? Has this been computed? In a more in-depth study I recommend to show both S and N along with U, for the same sources considered in this study.**

We acknowledge that, especially in the far-field, geoid changes and global mean mass changes can be as important as vertical land motion. However, those signals are a part of the tide gauge observations that researchers want to preserve. It is vertical land motion that often represents a nuisance signal, which is the reason why we have decided to make it the object of the current study.

**L111. The coseismic displacement can be also modelled globally (see http://onlinelibrary.wiley.com/doi/10.1029/2003GL019347/full).**

We have added the suggested reference to Melini et al. (2004).

**L112. What is the signal identified therein? Unclear. Is the rate of solar motion driven by the ice sources considered? What is its amplitude and direction?**

Indeed, we do not specifically quantify the size and direction of the pole tide (now "earth rotational effects") driven by ice mass loss, because it is beyond the scope of this study.
We have clarified the sentence, which now reads "meaning that the decadal and secular signals contributing to vertical land motion as identified in this study are not considered".

**L130. I do not understand why the 'far field signature' is mentioned here. Viscosity also controls deformation in the near field.**

The far-field signature is the main object of this study. Instead of "controlled by bulk viscosity values" we now say "controlled by viscoelastic relaxation mostly taking place deep in the mantle". We agree that also the near field is controlled by viscosity, but near field relaxation is more sensitive to shallow mantle regions, where viscosity values could be much lower and provide significant responses even at decadal scales.

**See http://journals.fcla.edu/jcr/article/view/80095/77355 for advice on how hyphenate "sea level".**

Thank you for the reference, we have harmonized hyphenation of "sea level".